# Lysosomal Stress in Cardiovascular Diseases: Therapeutic Potential of Cardiovascular Drugs and Future Directions

**DOI:** 10.3390/biomedicines13051053

**Published:** 2025-04-27

**Authors:** Toshiki Otoda, Ken-ichi Aihara, Tadateru Takayama

**Affiliations:** 1Division of General Medicine, Department of Internal Medicine, Nihon University School of Medicine, 30-1 Oyaguchikamicho, Itabashi, Tokyo 173-8610, Japan; takayama.tadateru@nihon-u.ac.jp; 2Department of Community Medicine and Medical Science, Tokushima University Graduate School of Biomedical Sciences, 3-18-15, Kuramoto-cho, Tokushima 770-8503, Japan; aihara@tokushima-u.ac.jp

**Keywords:** lysosomal stress, statins, transcription factor EB, glutaminase 1, senescence, NLRP3 inflammasome, SGLT2 inhibitors, regulatory complex, trehalose

## Abstract

Lysosomal dysfunction has emerged as a central contributor to the pathogenesis of cardiovascular diseases (CVDs), particularly due to its involvement in chronic inflammation, lipid dysregulation, and oxidative stress. This review highlights the multifaceted roles of lysosomes in CVD pathophysiology, focusing on key mechanisms such as NLRP3 inflammasome activation, TFEB-mediated autophagy regulation, ferroptosis, and the role of apolipoprotein M (ApoM) in preserving lysosomal integrity. Additionally, we discuss how impaired lysosomal acidification, mediated by V-ATPase, contributes to lipid-induced cardiac dysfunction. Therapeutically, several pharmacological agents, such as statins, SGLT2 inhibitors, TRPML1 agonists, resveratrol, curcumin, and ferroptosis modulators (e.g., GLS1 activators and icariin), have demonstrated promise in restoring lysosomal function, enhancing autophagic flux, and reducing inflammatory and oxidative injury in both experimental models and early clinical settings. However, key challenges remain, including limitations in drug delivery systems, the absence of lysosome-specific biomarkers, and insufficient clinical validation of these strategies. Future research should prioritize the development of reliable diagnostic tools for lysosomal dysfunction, the optimization of targeted drug delivery, and large-scale clinical trials to validate therapeutic efficacy. Incorporating lysosome-modulating approaches into standard cardiovascular care may offer a new precision medicine paradigm for managing CVD progression.

## 1. Introduction

Cardiovascular diseases (CVDs) remain the leading cause of morbidity and mortality worldwide and are primarily driven by metabolic disorders, such as obesity, type 2 diabetes mellitus, and atherosclerosis [1]. These conditions accelerate CVD progression through chronic inflammation, oxidative stress, and metabolic dysregulation, ultimately resulting in complications such as chronic cardiomyopathy, pressure overload, and heart failure (HF) [2]. Metabolic dysfunction contributes to various forms of organelle stress, including endoplasmic reticulum stress, mitochondrial dysfunction, and lysosomal stress, all of which collectively disrupt cellular homeostasis and promote vascular inflammation [3,4,5]. Ref. [6] notes the disruption of lysosomal homeostasis, and Ref. [7] further emphasizes the promotion of vascular inflammation and atherosclerosis, which are crucial for the pathogenesis of cardiovascular complications.

Lysosomes are essential organelles responsible for the degradation and recycling of cellular waste via autophagy and endocytosis [8]. However, emerging evidence has expanded the understanding of lysosomes beyond their conventional roles, revealing them as dynamic signaling hubs regulating various cellular processes, including the mechanistic target of the mammalian target of rapamycin complex 1 (mTORC1), 5′ adenosine monophosphate-activated protein kinase (AMPK), and inflammasome signaling pathways [9,10]. Additionally, lysosomes contribute to extracellular processes, such as microcirculation, through secretory lysosomes, underscoring their systemic impact on cellular and tissue health.

Lysosomes are frequently subjected to damage due to various internal and external factors [11]. When damaged, they release protons and cathepsins, leading to cell death and inflammation [11]. Recent studies have shown that cells possess a collective defense mechanism known as the lysosomal damage response to mitigate the effects of lysosomal injury. This response includes lysophagy, a form of selective autophagy; membrane repair pathways mediated by endosomal sorting complexes required for protein transport; and biosynthetic pathways regulated by transcription factor EB (TFEB). These mechanisms function synergistically to maintain lysosomal homeostasis [12].

Lysosomal dysfunction has been implicated in rare genetic conditions, such as lysosomal storage diseases, and prevalent age-related disorders, including cardiovascular and neurodegenerative diseases [13,14]. The concept of “lysosomal stress” has recently emerged, emphasizing its critical role in driving chronic inflammation, oxidative stress, and metabolic dysregulation, which are significant contributors to CVD progression [15]. This emerging evidence suggests that lysosomes are promising therapeutic targets for addressing the mechanisms underlying CVDs.

Recent advances in cardiovascular drugs have led to the investigation of strategies that target lysosomal pathways to mitigate inflammation, oxidative stress, and metabolic dysfunction [16]. By modulating lysosomal activity, these therapies aim to restore cellular homeostasis, reduce the progression of atherosclerosis, and stabilize vulnerable plaques. This review explores the role of lysosomal dysfunction in CVDs and discusses emerging cardiovascular drugs that leverage lysosome-targeted approaches to improve clinical outcomes.

## 2. Lysosomal Dysfunction, NLRP3 Inflammasome, and CVDs: Mechanisms and Therapeutic Insights

Lysosomal dysfunction, particularly lysosomal rupture, is vital for the pathogenesis of CVDs. Lysosomal rupture involves the complete loss of lysosomal membrane integrity, resulting in the uncontrolled release of cathepsins into the cytosol [17]. This process triggers potent inflammatory responses, including the activation of the NOD, LRR, and pyrin domain-containing protein 3 (NLRP3) inflammasome, which is a crucial mediator of chronic inflammation in CVDs (Figure 1). Latz et al. first demonstrated that cholesterol crystals internalized by macrophages in atherosclerotic plaques induce lysosomal rupture, releasing cathepsin B and driving NLRP3 activation, leading to sustained vascular inflammation [18].

Similarly, Karasawa et al. showed that saturated fatty acids, such as palmitic acid, induce intracellular crystal formation, causing lysosomal damage and activating the NLRP3 inflammasome in macrophages [19]. Conversely, unsaturated fatty acids, such as oleic acid, prevent this process by inhibiting crystal formation, underscoring the therapeutic significance of maintaining the balance between saturated and unsaturated fatty acids [19].

In contrast, lysosomal membrane permeabilization (LMP) represents a partial and controlled disruption of lysosomal membranes that regulates NLRP3 inflammasome activity under specific conditions [20]. For example, in human vascular smooth muscle cells, LMP acts as an activation signal linking lysosomal dysfunction to vascular inflammation [21]. These insights highlight the potential of targeting lysosomal integrity and inflammasome regulation as therapeutic strategies for CVD.

**Figure 1 biomedicines-13-01053-f001:**
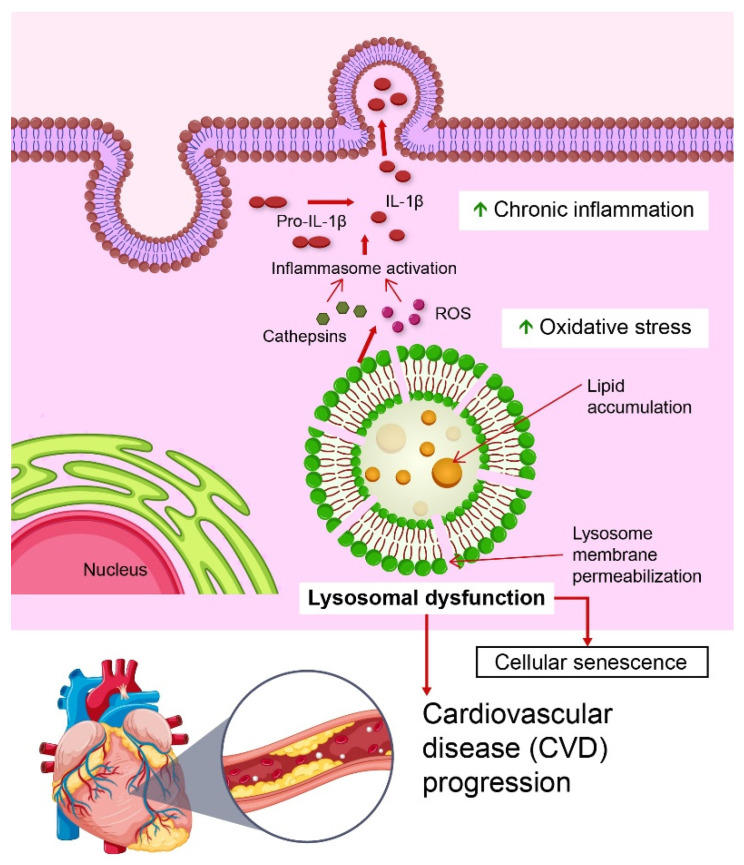
Role of lysosomal dysfunction in the progression of cardiovascular disease (CVD). Lysosomal dysfunction, characterized by lysosomal membrane permeabilization and lipid accumulation, leads to the release of cathepsins and reactive oxygen species (ROS). These factors activate inflammasomes, promoting the cleavage of pro-IL-1β into its active form, IL-1β, which triggers chronic inflammation. In addition, oxidative stress exacerbates lysosomal damage [22]. These processes collectively drive cellular senescence and contribute to CVD progression. This figure highlights the interplay between lysosomal dysfunction, chronic inflammation, and oxidative stress in the pathogenesis of CVD.

### 2.1. Cardiovascular Drugs Targeting Lysosomal Stress and NLRP3 Inflammasome Activation

Several cardiovascular drugs have demonstrated the potential to mitigate lysosomal stress and associated inflammation by restoring lysosomal function and suppressing NLRP3 inflammasome activation.

#### 2.1.1. Statins

In addition to their lipid-lowering effects, HMG-CoA reductase inhibitors (statins) have been shown to suppress NLRP3 inflammasome activation through PPAR-γ activation, highlighting their potential anti-inflammatory benefits beyond cholesterol reduction [23]. By inducing autophagy, simvastatin promotes the clearance of damaged lysosomes, enhances lysosomal biogenesis, preserves endothelial barrier integrity, and reduces cardiovascular risk in the context of obesity and diabetes [24].

#### 2.1.2. SGLT2 Inhibitors

Dapagliflozin, a sodium–glucose co-transporter-2 (SGLT2) inhibitor, has demonstrated cardioprotective effects beyond glucose lowering [25,26]. In a mouse model of myocardial ischemia/reperfusion (I/R) injury, dapagliflozin reduced infarct size, cardiac damage markers, and inflammation by suppressing NLRP3 inflammasome activation. These effects are mediated by enhanced autophagy and improved lysosomal function, which promotes NLRP3 clearance [27]. In primary cardiomyocytes, dapagliflozin reduced intracellular Ca^2^⁺ and Na⁺ levels, further supporting its role in protecting against I/R injury.

#### 2.1.3. Vitamin E

In vivo, Vitamin E predominantly exists in its free form, which enables it to exert antioxidative effects more efficiently than its esterified form [28]. Clinical studies indicate that a minimum of 100 IU of Vitamin E is required to inhibit LDL oxidation [29], a key process in the development of atherosclerosis and CVD. Several clinical trials, including the Cambridge Heart Antioxidant Study (400–800 IU/day) and the SPACE trial (800 IU/day), have reported the cardioprotective effects of Vitamin E [30,31]. However, other trials, such as GISSI [32], HOPE [33], and PPP [34], have failed to confirm its efficacy, contributing to an ongoing debate regarding the role of oxidative stress in CVD. Levy et al. suggested that individuals with reduced antioxidative capacity may derive greater benefits from Vitamin E supplementation [35].

### 2.2. Regulatory Complex and CVD

The regulatory complex of lysosomal membrane proteins plays a role in linking lysosomal metabolic signaling pathways with immune response regulation mechanisms [36]. The regulator complex, located on the lysosomal membrane, is composed of Lamtor2 (p14), Lamtor3 (MP10), Lamtor4 (p10), and Lamtor5 (HBXIP), which are encapsulated by Lamtor1 (p18) [37]. Among these components, Lamtor1 is indispensable for maintaining the overall stability of the regulatory complex. The regulator complex primarily functions in metabolic signaling by anchoring mTORC1 to the lysosomal membrane, integrating intracellular nutrient signals, and regulating protein and lipid synthesis and degradation. Kimura et al. reported that the Lamtor1-mTORC1 pathway transmits signals to liver X receptors (LXRs) based on the macrophage environment and nutrient status, thereby modulating the activation of M2 (anti-inflammatory) macrophages [38]. Hayama et al. demonstrated that the regulation of TFEB via the Lamtor1-mTORC1 pathway is involved in lysosomal autophagy and the production of inflammatory cytokines [39]. Recent advances, including the findings of Evavold et al., have indicated that the Lamtor1-mTORC1 pathway promotes the production of mitochondrial ROS and subsequent pyroptosis through gasdermin D (GSDMD) pore formation. These studies underscore the role of the regulator complex in linking lysosomal metabolic signaling pathways with immune response regulation mechanisms. Emerging research has highlighted the regulatory complex as a mediator of NLRP3 inflammasome activation. Lamtor1, a regulatory complex component, interacts with NLRP3 and histone deacetylase 6 (HDAC6) to modulate inflammasome activity. Lamtor1 deletion in macrophages significantly reduced NLRP3 activation, demonstrating its regulatory role in inflammation. Furthermore, DL-all-rac-α-tocopherol, a synthetic form of Vitamin E, inhibits the interaction between Lamtor1 and HDAC6, reducing NLRP3 activation and alleviating inflammation in gouty arthritis models [40]. Vitamin E has also been associated with a reduction in cardiovascular events [30,31,35]. However, while evidence supports the role of DL-all-rac-α-tocopherol in inhibiting Lamtor1-HDAC6 interactions and reducing NLRP3 activation under inflammatory conditions, its effects on cardiovascular disease remain inconclusive. Specifically, it has not been definitively established whether Vitamin E exerts the same inhibitory effects in cardiovascular pathology. The direct link between Vitamin E-mediated inhibition of Lamtor1-HDAC6 interactions, reduced NLRP3 activation, and the prevention of cardiovascular events remains unverified. Future studies are needed to clarify whether these mechanisms contribute to the cardioprotective effects of Vitamin E and its potential as a therapeutic target for cardiovascular disease.

### 2.3. Uric Acid, Lysosomal Dysfunction, and NLRP3 Inflammasome Activation in Cardiovascular Disease

Uric acid (UA) crystals function as damage-associated molecular patterns (DAMPs), triggering immune activation and inflammation [41]. Although the pro-inflammatory effects of UA crystals have been extensively studied, emerging evidence suggests that soluble uric acid (sUA) may also contribute to organ damage and chronic inflammation, particularly in renal complications and CVD [42]. Studies have indicated that sUA promotes the production of inflammasome-related molecules, leading to cardiomyocyte apoptosis and cardiovascular dysfunction [43]. Hyperuricemia, characterized by elevated serum UA levels, is strongly associated with gout, chronic kidney disease, and cardiovascular complications [44]. UA crystals are well-documented activators of the NLRP3 inflammasome, primarily through frustrated phagocytosis or lysosomal rupture, which results in the release of cathepsins and ATP into the cytoplasm [45]. These lysosomal contents act as danger signals, amplifying inflammasome activation, inflammation, and subsequent tissue injury [46]. Although the precise molecular pathways remain unclear, therapeutic strategies aimed at preserving lysosomal integrity or inhibiting the release of cathepsins and ATP, such as probenecid, which has been investigated in brain reperfusion injury, have shown promise in suppressing NLRP3 inflammasome activation and mitigating tissue damage [47]. Notably, sUA appears to activate NLRP3 through a mechanism independent of lysosomal damage, which distinguishes it from inflammation induced by UA crystals [48]. The precise molecular pathways by which sUA and UA crystals drive NLRP3 activation require further investigation [49]. A more thorough understanding of these mechanisms could pave the way for targeted therapies that not only preserve lysosomal function and regulate inflammasome activity but also modulate UA metabolism, ultimately reducing cardiovascular and renal complications associated with hyperuricemia [50].

### 2.4. PCSK9 Inhibition and Its Potential Impact on Lysosomal Function

Although PCSK9 inhibitors have been widely studied for their role in cholesterol regulation and atherosclerosis management, their effects on lysosomal function remain largely unexplored [51]. Various forms of PCSK9 inhibitors, including monoclonal antibodies (mAbs), small interfering RNAs (siRNAs), and vaccines, have demonstrated clinical efficacy in lowering LDL cholesterol levels and reducing cardiovascular risk [52,53,54]. Recent findings suggest that PCSK9 interacts with the NLRP3 inflammasome, potentially influencing the inflammatory processes involved in atherosclerosis progression [55,56]. However, whether PCSK9 inhibition affects lysosomal integrity, autophagy regulation, or lipid processing within lysosomes is yet to be fully established [57,58]. Given the fundamental role of lysosomes in cholesterol metabolism and immune signaling, further investigation is required to determine whether PCSK9 inhibition modulates lysosomal stability, autophagic flux, or NLRP3 inflammasome activation via lysosomal pathways [59,60]. Clarifying these mechanisms could lead to novel therapeutic strategies that integrate PCSK9 inhibitors with lysosome-targeting therapies to optimize the treatment of atherosclerosis, metabolic disorders, and inflammatory CVDs [61,62].

## 3. TFEB: A Key Player in Lysosomal Stress and Cardiovascular Therapy

Lysosomal dysfunction plays a critical role in the progression of CVDs by contributing to chronic inflammation, oxidative stress, and impaired metabolic regulation. Recent research has identified TFEB as a central regulator of lysosomal homeostasis, making it a key target for therapeutic intervention in CVDs [63]. TFEB functions as a master transcriptional regulator, coordinating lysosomal biogenesis, autophagy, and cellular adaptation to metabolic stress [64]. Given its essential role in maintaining cellular quality control, TFEB activation is increasingly being recognized as a promising strategy for mitigating cardiovascular complications.

One of the primary mechanisms by which TFEB exerts its protective effects is through the enhancement of lysosomal biogenesis and function [65]. By upregulating genes involved in lysosomal stability and acidification, TFEB ensures the efficient degradation of cellular debris and prevents lysosomal rupture, which is a key driver of inflammatory responses in atherosclerosis and HF. Furthermore, TFEB activation supports autophagy, which is essential for the removal of damaged mitochondria and lipid-laden foam cells, thereby reducing vascular inflammation and plaque instability [66]. Recent studies have suggested that TFEB also plays a crucial role in mitigating oxidative stress, which is a major contributor to endothelial dysfunction and cardiomyocyte apoptosis [67,68,69]. By regulating lysosomal repair pathways, TFEB helps maintain redox balance, which is essential for preserving cardiovascular function under conditions of metabolic and hemodynamic stress. As the molecular pathways governing TFEB activation become better understood, there is a growing interest in leveraging this transcription factor in cardiovascular therapy [70,71]. The following sections discuss the involvement of TFEB in atherosclerosis, its role in vascular smooth muscle cell stability, and the therapeutic potential of pharmacological agents in enhancing TFEB activity.

### 3.1. Role of TFEB in Atherosclerosis and Endothelial Damage

Emanuel et al. [19] demonstrated that oxidized low-density lipoproteins and cholesterol crystals induce lysosomal dysfunction in macrophages. TFEB overexpression in these cells restores lysosomal function, promotes autophagy, and reduces inflammation, highlighting its therapeutic potential in vascular inflammation and atherosclerosis [72].

### 3.2. TFEB in Vascular Smooth Muscle Cells and Plaque Stability

Vascular smooth muscle cells (VSMCs) are significant contributors to plaque stability, partly through the production of cystathionine gamma-lyase (CTH), which generates hydrogen sulfide (H_2_S), a protective gasotransmitter in atherosclerosis. Reduced CTH expression in VSMCs has been associated with impaired autophagy, lysosomal dysfunction, and increased plaque vulnerability [25]. Experimental studies have demonstrated that *Cth* deletion in VSMCs exacerbates lysosomal and autophagic deficits, increases apoptosis, and reduces collagen secretion, leading to plaque instability [73]. H_2_S donors have been shown to rescue these effects by activating TFEB, a master regulator of autophagy and lysosomal function. H_2_S sulfhydrates TFEB at Cys212 and promotes its nuclear translocation and transcriptional activation of lysosomal and autophagy-related genes, such as *ATG9A* and *LAPTM5*. This activation restores lysosomal function, enhances autophagic flux, supports collagen production, and stabilizes plaques. These findings underscore the significance of the CTH-H2S-TFEB axis in maintaining lysosomal integrity and autophagy in VSMCs, offering promising therapeutic and biomarker potential for vulnerable plaques.

### 3.3. Cardiovascular Drugs Enhancing TFEB Activity

#### 3.3.1. Statins

Simvastatin: This promotes TFEB activation by inhibiting mTORC1, activating AMPK, enhancing autophagy, and reducing lipid accumulation in atherosclerotic models [74].

Atorvastatin: It increases TFEB protein levels in cardiac tissue and improves lysosomal function in doxorubicin-induced cardiomyopathy, as indicated by an enhanced lysosomal membrane protein (LAMP)2/tubule-associated protein 1 light chain 3 beta (LC3B) ratio [75].

#### 3.3.2. TRPML1 Agonists

The lysosomal Ca^2^⁺ channel, transient receptor potential mucolipin1 (TRPML1), is selectively degraded in response to lysosomal stress, contributing to the regulation of lysosomal function and size [76]. TRPML1 facilitates calcium release during lysosomal stress, activates TFEB, and promotes autophagy. Small-molecule TRPML1 agonists prevent mitochondrial fragmentation, reduce ROS, and restore redox balance in endothelial cells exposed to saturated fatty acids [77]. TRPML1 overexpression enhanced resistance to oxidative damage, whereas its silencing negated TFEB-mediated protective effects, suggesting a complementary role for TRPML1 agonists in statin therapy.

#### 3.3.3. Trehalose

Trehalose, a natural disaccharide, has demonstrated efficacy as a pharmacological inducer of autophagy and activator of TFEB. It restores lysosomal function in macrophages, reduces polyubiquitinated protein aggregates, and suppresses apoptosis and IL-1β secretion, significantly decreasing plaque burden without affecting systemic cholesterol levels [78]. Trehalose elevates lysosomal pH, inactivates mTORC1, and promotes TFEB nuclear translocation, thereby enhancing lysosomal biogenesis and autophagy [79]. Additionally, in myocardial infarction models, trehalose reduced ventricular remodeling, apoptosis, and fibrosis by enhancing autophagic flux and TFEB activation [80]. Its diminished effects in autophagy-deficient models emphasize its reliance on lysosomal pathways, making it a promising therapeutic candidate for CVDs associated with lysosomal dysfunction.

#### 3.3.4. Resveratrol (RSV)

RSV, a naturally occurring polyphenol found in fruits and nuts, has been shown to reduce intracellular lipid accumulation, although the precise molecular mechanism remains unclear [81]. Recent studies have indicated that RSV stimulates endoplasmic reticulum (ER)-Ca^2^⁺ signaling, which in turn activates protein phosphatase 2A (PP2A) [82]. This activation facilitates TFEB dephosphorylation, allowing its translocation to the nucleus and promoting the expression of key genes involved in autophagy and lysosomal biogenesis [83]. Notably, genetic suppression of TFEB significantly diminishes RSV-induced lipid degradation, confirming its central role in metabolic effects [82]. These findings suggest that RSV enhances lysosomal function and autophagic lipid clearance through TFEB activation, thus presenting a promising therapeutic approach for metabolic diseases associated with lipid dysregulation [84].

#### 3.3.5. Curcumin (Cur)

Cur, a polyphenol derived from *Curcuma longa*, has been shown to mitigate atherosclerosis by promoting TFEB activation, enhancing autophagy, and reducing inflammation [85]. It facilitates the nuclear translocation of TFEB and improves lysosomal biogenesis and lipid catabolism in foam cells, thereby preventing excessive lipid accumulation. Beyond its role in autophagy, curcumin suppresses inflammation by inhibiting the P300-BRD4 axis, which is involved in the transcriptional activation of pro-inflammatory genes in atherosclerosis. This dual mechanism not only enhances lipid clearance but also reduces oxidative stress and inflammatory cytokine production. In *ApoE*-deficient mouse models, curcumin administration reduces atherosclerotic plaque formation, improves vascular integrity, and restores autophagic flux. These effects depend on TFEB activation, highlighting the potential of curcumin as a therapeutic agent for lysosomal dysfunction in CVDs.

#### 3.3.6. Traditional Chinese Medicine: Dehydroandrographolide (DA)

DA, an active component of *Andrographis paniculata*, mitigates doxorubicin-induced cardiotoxicity by enhancing lysosomal function and autophagic flux. DA inhibits mTOR activity and promotes TFEB nuclear translocation [86]. Although these findings highlight the potential of DA in repairing lysosomal dysfunction in CVDs, its application remains limited to experimental models.

### 3.4. Eicosapentaenoic Acid (EPA) and Lysosomal Homeostasis

Large-scale randomized clinical trials, including the REDUCE-IT [87] and STRENGTH [88] trials, have investigated the cardiovascular benefits of high-dose omega-3 polyunsaturated fatty acids (PUFAs). Notably, REDUCE-IT reported a 25% reduction in cardiovascular events, whereas STRENGTH showed only a 1% risk reduction, leading to conflicting interpretations regarding whether EPA truly reduced cardiovascular risk. Subsequent cohort studies have suggested that differences in the comparator oils used in these trials may partially explain the disparity in outcomes [89]; however, the overall efficacy of EPA remains uncertain.

Beyond its established roles in lipid metabolism, inflammation reduction, and endothelial function improvement, emerging evidence suggests that EPA may also influence lysosomal function, which is a key component of cellular homeostasis. Lysosomes facilitate the degradation of damaged organelles and proteins via autophagy, a process essential for cardiovascular protection. Lysosomal dysfunction contributes to CVDs, including ischemia/reperfusion injury and atherosclerosis, by impairing cellular clearance mechanisms and exacerbating oxidative stress. Several studies have indicated that EPA promotes autophagy, facilitating adaptive cellular responses that counteract oxidative stress-induced cardiomyocyte apoptosis [90]. However, whether this effect is mediated through the direct modulation of lysosomal pathways remains unclear. Given the fundamental role of lysosomes in cholesterol metabolism, inflammatory regulation, and cellular clearance, further research is required to determine the precise effects of EPA on lysosomal integrity, autophagic flux, and metabolic balance. Future investigations should explore whether EPA influences key lysosomal regulators, such as TFEB or TRPML1, to provide valuable insights into its role in lysosome-targeted therapies. A deeper understanding of the interactions between EPA and lysosomal biology may lead to novel therapeutic strategies for CVDs associated with lysosomal dysfunction and impaired autophagy.

## 4. Ferroptosis in Cardiovascular Disease: From Plaque Destabilization to Myocardial Injury

Ferroptosis, an iron-dependent form of cell death, plays a crucial role in the destabilization of atherosclerotic plaques and the progression of CVDs. Metabolic disorders, such as obesity, diabetes, and atherosclerosis, promote ferroptosis by increasing oxidative stress, lipid peroxidation, and lysosomal dysfunction, thereby exacerbating cardiovascular risk [91,92,93,94]. Lysosomal dysfunction contributes to this process by releasing iron and ROS, which amplifies ferroptosis in VSMCs and increases plaque vulnerability. Recent studies have further highlighted metabolic reprogramming in unstable plaques with intraplaque hemorrhage (IPH+), revealing impaired glutamine-to-glutamate conversion and reduced glutamate levels. These metabolic alterations are associated with an increased presence of macrophages and a pro-inflammatory microenvironment. The interplay between metabolic dysfunction and ferroptosis exacerbates plaque instability, underscoring the therapeutic potential of targeting lysosomal dysfunction and metabolic pathways in CVDs [95].

A recent study revealed that the Yes-associated protein 1 (YAP1)/glutaminase 1 (GLS1) axis is a significant regulator of ferroptosis in VSMCs [96]. GLS1, an enzyme essential for glutamine metabolism, enhances the production of glutamate and glutathione, which are critical for maintaining redox balance and preventing lipid peroxidation. YAP1 signaling activates GLS1, mitigates ferroptosis, stabilizes VSMCs, and reinforces atherosclerotic plaques. Conversely, GLS1 inhibition exacerbates oxidative stress and plaque instability, suggesting that enhancing GLS1 activity could serve as a novel therapeutic strategy to counteract ferroptosis in CVDs.

### 4.1. Cardiovascular Drugs Targeting Ferroptosis and Lysosomal Stress

#### 4.1.1. GLS1 Activators

Pharmacological agents targeting GLS1 or promoting glutathione synthesis complement statins by directly reducing ferroptosis in VSMCs and stabilizing plaques [96]. Combining statins with GLS1 activators has potential synergistic effects on plaque stability and overall cardiovascular outcomes.

#### 4.1.2. Icariin

Icariin, derived from *Epimedium brevicornum Maxim*., has demonstrated protective effects against ferroptosis in vascular endothelial cells exposed to oxidized low-density lipoproteins [97]. It reduces ROS levels, promotes TFEB nuclear translocation, enhances autophagosome–lysosome fusion, increases autophagy, and decreases ferroptosis. In high-fat diet-fed ApoE-deficient mice, icariin alleviated atherosclerotic lesions, demonstrating its potential as a candidate for CVD treatment by targeting lysosomal pathways and ferroptosis.

#### 4.1.3. Cur

Curcumin (Cur), a polyphenol derived from *Curcuma longa*, protects against myocardial ischemia/reperfusion injury (MIRI) by inhibiting autophagy-dependent ferroptosis [98]. In a Sprague Dawley rat model of MIRI and an H9c2 cardiomyocyte anoxia/reoxygenation (A/R) model, Cur was found to activate the Sirt1/AKT/FoxO3a pathway, preventing FoxO3a from inducing autophagy-related genes (LC3II and ATG12) and thereby reducing ferroptosis. Additionally, Cur reduced iron overload, lipid peroxidation (measured by malondialdehyde), and ROS while restoring glutathione (GSH) and superoxide dismutase levels.

#### 4.1.4. Other Therapeutic Approaches

Treatment with Ferrostatin-1 (a ferroptosis inhibitor) and dexrazoxane (an iron chelator) prevented the increase in cardiac biomarker levels induced by ischemia–reperfusion injury and reduced myocardial infarction size [99]. Furthermore, the inhibition of glutaminolysis, a crucial pathway in ferroptosis, attenuates ischemia–reperfusion-related cardiac damage [100].

## 5. Future Challenges in Bridging Experimental and Clinical Research on Lysosomal Dysfunction

We summarize the therapeutic strategies targeting lysosomal dysfunction in cardiovascular diseases in Figure 2, which highlights key pharmacological agents and their corresponding molecular targets. Lysosomal dysfunction, NLRP3 inflammasome activation, and ferroptosis have emerged as critical mechanisms in the pathogenesis of CVDs, including atherosclerosis, HF, and metabolic syndrome [18,19,41,101,102]. Although significant progress has been made in understanding these pathways, several challenges must be addressed to translate these findings into effective clinical therapies.

### 5.1. Advancing Lysosome-Targeted Therapies

Therapeutic strategies aimed at modulating lysosomal function, including TFEB activators, TRPML1 agonists, and autophagy enhancers, have yielded promising preclinical results [5,83]. However, their long-term safety and clinical efficacy require further validation. Although statins and SGLT2 inhibitors have been widely used in CVD treatment and have shown effects on lysosomal function, their direct impact on lysosomal stress pathways remains unclear [24,27]. Future research should focus on refining lysosome-targeted drugs, optimizing their bioavailability, and evaluating their effects in large-scale clinical trials.

Additionally, compounds such as trehalose, resveratrol, and curcumin have been found to enhance lysosomal function through TFEB activation. However, their low bioavailability and metabolic instability limit their therapeutic potential. Advanced drug delivery systems, including nanoparticle-based formulations, can overcome these challenges and enhance clinical applicability [103,104].

### 5.2. Overcoming Challenges in NLRP3 Inflammasome Inhibition

Inflammation is a key driver of CVD progression, and NLRP3 inflammasome activation plays a central role in this process. Pharmacological inhibitors of the NLRP3 inflammasome, such as statins, SGLT2 inhibitors, and Vitamin E, have demonstrated anti-inflammatory and cardioprotective effects. However, the precise mechanisms underlying the modulation of lysosomal function remain unclear. Further research is needed to determine whether lysosomal integrity directly regulates inflammasome activation in cardiovascular pathology and how these interventions can be optimized for long-term use.

Emerging evidence suggests that UA crystals can activate the NLRP3 inflammasome through lysosomal rupture, contributing to tissue damage in hyperuricemia-related CVDs [41,42,43]. Future studies should explore how targeting UA metabolism and lysosomal integrity can reduce inflammation and improve CVD outcomes.

### 5.3. Addressing the Role of Ferroptosis in Cardiovascular Disease

Ferroptosis, a form of iron-dependent programmed cell death, is implicated in atherosclerotic plaque instability and cardiomyocyte injury [99]. Therapeutic interventions targeting ferroptosis, such as GLS1 activators and ferroptosis inhibitors, such as icariin, have shown potential in preclinical models. However, its clinical relevance remains unclear. Further studies should aim to clarify the interplay between lysosomal iron metabolism, ferroptosis, and oxidative stress in CVDs.

Strategies that integrate ferroptosis modulation with lysosome-targeted therapies may provide synergistic benefits in stabilizing atherosclerotic plaques and mitigating ischemia–reperfusion injury. The development of biomarkers to monitor ferroptosis in patients with cardiovascular risk factors is critical for assessing therapeutic efficacy.

### 5.4. Future Perspectives and Translational Research

To successfully implement lysosome-targeted therapies in CVD treatment, future research should prioritize the following:Clinical Trials and Drug Optimization: Expanding human trials for lysosome-modulating drugs, including TFEB activators, TRPML1 agonists, and ferroptosis inhibitors, is crucial for clinical translation.Personalized Medicine Approaches: Identifying patient subgroups with heightened lysosomal stress or inflammasome activation could allow for targeted interventions tailored to specific cardiovascular conditions.Biomarker Development: Establishing reliable biomarkers for lysosomal dysfunction, NLRP3 activation, and ferroptosis will aid in disease diagnosis, treatment monitoring, and therapy selection.

By addressing these challenges, lysosome-targeted interventions have the potential to revolutionize CVD treatment, offering a novel approach for mitigating chronic inflammation, oxidative stress, and metabolic dysfunction associated with cardiovascular pathology.

Key therapeutic strategies include the following:NLRP3 Inflammasome Inhibition: Statins, SGLT2 inhibitors, and Vitamin E suppress NLRP3 activation, thereby reducing inflammation.TFEB Activation: Statins, TRPML1 agonists, trehalose, resveratrol, curcumin, and dehydroandrographolide (DA) enhance lysosomal biogenesis and autophagic clearance.Ferroptosis Modulation: GLS1 activators, icariin, and curcumin, regulate iron-dependent cell death, thereby protecting against cardiovascular damage.Lysosomal Repair Mechanisms: TRPML1 activation and lysophagy modulation stabilize lysosomal membranes and restore function.

By restoring lysosomal homeostasis, these strategies may reduce oxidative stress and improve cardiovascular outcomes. Further clinical validation is required.

## 6. Lysosomal Dysfunction and Cellular Senescence: Implications for Cardiovascular Health

Cellular senescence is a state of irreversible cell cycle arrest induced by stressors such as DNA damage, oxidative stress, and oncogenic signals [105]. Despite their inability to divide, senescent cells remain metabolically active and secrete various inflammatory and matrix-degrading factors, referred to as the senescence-associated secretory phenotype (SASP) [106,107]. The SASP promotes chronic inflammation and tissue remodeling, significantly contributing to the development of CVDs and obesity-related metabolic disorders [108].

Recent evidence suggests that eliminating senescent cells alleviates age-related conditions, including atherosclerosis and chronic kidney disease, while extending the health span of animal models [109]. For example, selective clearance of senescent cells in transgenic mice reduces atherosclerotic plaque size, improves cardiac function, and mitigates systemic inflammation. These findings have prompted the development of senotherapy, a therapeutic approach aimed at targeting senescent cells to treat or prevent aging-related diseases. Several of these senotherapeutics have advanced to clinical trials, providing potentially promising treatment options for age-related conditions and disorders, including atherosclerosis, cancer, diabetes, neurodegenerative diseases, CVDs, and chronic kidney disease [110].

### 6.1. Lysosomal Dysfunction in Cellular Senescence

Lysosomal dysfunction is a hallmark of cellular senescence and is characterized by the accumulation of cholesterol metabolites in the lysosomal limiting membrane. This forms cholesterol-rich microdomains, which act as platforms for mTORC1 signaling. Sustained mTORC1 activation in these domains has been associated with the persistent secretion of inflammatory SASP factors [14]. Such lysosomal alterations exacerbate the inflammatory milieu and impair autophagic flux, thereby contributing to metabolic and cardiovascular pathologies.

In addition, our findings demonstrated that the administration of SGLT2 inhibitors for four months significantly reduced serum inflammatory biomarkers, including the C-reactive protein and tumor necrosis factor receptor, in patients with type 2 diabetes [111]. This reduction was accompanied by a notable decrease in urinary albumin excretion, independent of improvements in hyperglycemia, obesity, hypertension, or hyperuricemia [111]. Although lysosomal function was not directly assessed, this study highlights the anti-inflammatory potential of SGLT2 inhibitors in mitigating metabolic and cardiovascular complications.

### 6.2. Cardiovascular Drugs Targeting Senescence via Lysosomal Pathways

Emerging cardiovascular drugs leverage the unique metabolic vulnerabilities of senescent cells to improve cardiovascular outcomes. Some examples are as follows:Statins: These are used to lower lipid levels. Statins have demonstrated the ability to inhibit mTORC1 activation by reducing intracellular cholesterol levels. This indirect modulation of SASP secretion may help mitigate chronic atherosclerotic inflammation.SGLT2 Inhibitors: By activating autophagic flux, SGLT2 inhibitors restore lysosomal function, reduce oxidative stress, and enhance cellular resilience against inflammation, offering potential benefits to patients with CVDs and diabetes [112].mTOR Inhibitors: Drugs such as rapamycin directly inhibit mTORC1 activity, reduce SASP secretion, and promote autophagy and lysosomal biogenesis, making them significant candidates for CVD treatment.

### 6.3. Targeting GLS1 for Cardiovascular Benefits

Recent studies have identified GLS1 as a novel target for immunotherapy [113]. In senescent cells, lysosomal damage induces intracellular acidification, stabilizes GLS1 mRNA, and enhances GLS1 protein expression. This mechanism increases ammonia production, neutralizes acidification, and supports the survival of senescent cells. GLS1 inhibition disrupts this compensatory mechanism, exacerbates intracellular acidification, and induces senescence. Preclinical studies have shown that GLS1 inhibitors can effectively clear senescent cells, reduce inflammation, and improve metabolic and cardiovascular health.

## 7. Lysosomal Stress and the Role of V-ATPase in Lipid-Induced Cardiac Dysfunction

Lysosomal stress has been identified as a critical factor in the pathogenesis of diabetic cardiomyopathy [114]. Lipid overload disrupts vacuolar H⁺-ATPase (V-ATPase) function, which is a key component of lysosomal and endosomal homeostasis, leading to endosomal deacidification. This disruption results in the aberrant trafficking of CD36, a lipid transporter, from the endosomes to the sarcolemma, exacerbating lipid accumulation, insulin resistance, and cardiac contractile dysfunction.

Wang et al. further demonstrated the therapeutic potential of specific amino acids (lysine, leucine, arginine) in mitigating lysosomal stress [115]. These amino acids reassembled V-ATPase, restored lysosomal and endosomal homeostasis, and reduced lipid accumulation in lipid-overloaded cardiomyocytes and high-fat diet-fed rats. The observed effects relied on the mTORC1-V-ATPase axis, adapter protein Raptor, and lysosomal amino acid transporter SLC38A9. These findings highlight lysosomal stress as a pivotal mechanism linking metabolic dysfunction and cardiovascular disease, underscoring the potential of targeting the V-ATPase-mTORC1 axis using specific amino acids as a novel therapeutic approach.

## 8. Apolipoprotein M (ApoM) and Lysosomal Function in Cardiovascular Disease

Clinical studies have indicated that reduced circulating ApoM is associated with increased mortality in patients with HF, independent of the HF subtype, ischemic heart disease, or HDL cholesterol levels [116,117]. ApoM, which binds to sphingosine-1-phosphate (S1P), is negatively affected by diabetes, particularly in HFpEFs. Recent findings suggest that anthracyclines, such as doxorubicin (Dox), reduce circulating ApoM levels in both mice and humans, whereas ApoM heterozygosity exacerbates DOX-induced cardiotoxicity [118]. Mechanistically, ApoM appears to sustain myocardial autophagic flux, protect against lysosomal injury, and mitigate DOX-induced cardiotoxicity. Studies using anthracycline-induced HF models have identified ApoM as a novel regulator of the autophagy–lysosomal pathway, highlighting its potential therapeutic role in HF management.

## 9. Conclusions

Integrating therapies targeting lysosomal dysfunction, cellular senescence, and ferroptosis into existing CVD treatment regimens has great potential for improving patient outcomes (Figure 3).

### 9.1. Pharmacological Approaches

Several pharmacological strategies are being explored to target lysosomal dysfunction in cardiovascular diseases. One key approach involves inhibiting the NLRP3 inflammasome, a major driver of inflammation and oxidative stress. Statins, SGLT2 inhibitors, and Vitamin E have demonstrated the ability to suppress NLRP3 activation, thereby reducing inflammatory responses and mitigating oxidative damage.

Another promising avenue focuses on activating TFEB, a transcription factor critical for lysosomal biogenesis and autophagy. Pharmacological agents such as statins, TRPML1 agonists, trehalose, resveratrol, curcumin, and dehydroandrographolide have been shown to enhance lysosomal function, thereby alleviating lipid overload and chronic inflammation.

Modulating ferroptosis, an iron-dependent form of cell death, is also emerging as a therapeutic target. Ferroptosis contributes to cardiovascular damage by disrupting lysosomal integrity and metabolic balance. Agents such as GLS1 activators, icariin, and curcumin have been investigated for their ability to regulate ferroptotic pathways and protect against cardiovascular dysfunction.

### 9.2. Emerging Therapeutic Targets

Beyond these established pharmacological approaches, novel strategies aimed at lysosomal dysfunction in cardiovascular diseases are gaining attention. One such area of interest is cellular senescence, where interventions targeting age-related lysosomal decline could help prevent cardiovascular aging and dysfunction.

Regulating lysosomal acidification through V-ATPase modulation has also been proposed as a strategy to restore autophagic flux and improve lipid degradation. Additionally, apolipoprotein M (ApoM) has been identified as a key player in maintaining endothelial lysosomal function through its influence on sphingosine-1-phosphate signaling, which is essential for vascular homeostasis.

Another potential target is the regulation of mTOR activity through ORP5-related pathways. The oxysterol-binding protein-related protein 5 (ORP5) has been implicated in mTORC1 regulation within lysosomes, influencing cardiovascular metabolism and contributing to conditions such as hypertrophy.

While these therapeutic strategies show promise, further clinical research is necessary to validate their efficacy and safety. Large-scale trials will be critical in determining their potential for integration into existing cardiovascular treatment regimens. A comprehensive table summarizing the key drugs discussed in this review, along with their mechanisms of action and clinical status, is provided below (Table 1).

Continued research in these areas is likely to yield innovative therapeutic strategies that address the underlying pathophysiology of CVDs and extend the health span of the aging population.

## 10. Future Directions in Lysosome-Targeted Cardiovascular Therapies

Lysosomal dysfunction plays a pivotal role in the pathogenesis of CVDs; however, its clinical significance remains underexplored. While preclinical studies have established that lysosomal stress contributes to inflammation, oxidative stress, and metabolic dysregulation in CVD, translating these findings into therapeutic applications remains challenging [9,11]. Future research should focus on refining lysosome-targeted strategies, integrating them into existing treatment regimens and developing clinical approaches for severe CVD, including HF [14,34]. Key future research directions in lysosome-targeted cardiovascular therapies are summarized in Table 2.

### 10.1. Advancing Lysosomal-Targeted Pharmacological Therapies

Current pharmacological strategies targeting lysosomal dysfunction in CVDs remain largely in the preclinical phase, with limited human data [24]. Existing therapies, such as statins and SGLT2 inhibitors, exert indirect protective effects by modulating lysosomal homeostasis and autophagy [24,27]. However, specific lysosome-targeted compounds, including TFEB activators (e.g., trehalose [78] and curcumin [85]) and TRPML1 agonists, require further clinical validation [76]. Future studies should assess their efficacy in stabilizing lysosomal function, reducing plaque instability, and mitigating ischemic injury. One major challenge is determining the optimal therapeutic window and dosage of these agents, as excessive autophagy activation can lead to maladaptive responses, including cell death [12]. Additionally, combination therapies integrating lysosomal modulators with established cardiovascular drugs should be explored to enhance treatment efficacy [61]. Given the heterogeneity of CVD, precision medicine approaches, such as identifying patient subgroups most likely to benefit from lysosomal-targeted interventions, should be prioritized [70].

### 10.2. Clinical Trials on Lysosome-Targeted Therapies in CVD

While lysosomal dysfunction is increasingly recognized as a critical factor in the pathogenesis of CVDs, clinical trials explicitly evaluating lysosomal function in patients with CVD remain absent. Several studies have indirectly explored pharmacological interventions with potential lysosomal modulatory effects, primarily through their roles in autophagy regulation, oxidative stress reduction, and inflammation control. However, these trials did not directly assess lysosomal integrity, autophagic flux, or lysosomal enzyme activity, highlighting a crucial gap in translational research.

Among the completed trials, NCT04061070 examined the impact of trehalose and polyphenols on oxidative stress and endothelial dysfunction in patients with peripheral arterial disease (PAD) and hypertension. Trehalose, a well-known autophagy enhancer, has been shown to protect lysosomal integrity and promote cellular clearance in preclinical studies. However, this study primarily focused on vascular function and did not include direct lysosomal assessments, leaving the precise impact on lysosomal homeostasis unclear.

### 10.3. Exploring Novel Biomarkers and Diagnostic Tools for Lysosomal Dysfunction

The assessment of lysosomal function in clinical practice remains limited due to a lack of reliable, non-invasive biomarkers [18]. Developing circulating biomarkers, such as lysosomal enzymes (cathepsins, LAMP2 [11]), TFEB activity, and lysosomal proteases, may enable the early detection of lysosomal dysfunction and provide insights into disease progression [14,63]. Moreover, advanced imaging techniques should be explored to visualize lysosomal activity in cardiovascular tissues. Emerging approaches, including PET tracers specific to lysosomal function [38] and MRI-based techniques [39], hold promise for assessing lysosomal health in vivo. By integrating these biomarkers with clinical risk stratification models, clinicians may be able to predict cardiovascular outcomes and tailor lysosomal-targeted therapies accordingly [40].

### 10.4. Translating Lysosome-Targeted Strategies into Clinical Practice

Lysosomal dysfunction has emerged as a key contributor to the progression of severe CVDs, particularly HF. Despite growing recognition of its importance, the direct clinical implications of lysosomal dysfunction in HF remain underexplored. Given that lysosomes regulate autophagy, metabolic homeostasis, and inflammatory signaling, their impairment can accelerate myocardial remodeling, fibrosis, and contractile dysfunction, thereby exacerbating HF progression. Understanding how lysosomal pathways influence these pathological mechanisms is crucial for the development of effective therapeutic strategies. Recent findings have identified mTORC1 activation in lysosomes as a key regulator of metabolic adaptations in the failing heart. Dysregulated mTORC1 signaling has been implicated in the pathogenesis of cardiac hypertrophy and metabolic inflexibility, both of which contribute to HF. In this context, oxysterol-binding protein-related protein 5 (ORP5) has been found to localize to lysosomal membranes, where it promotes mTORC1 activation, leading to enhanced anabolic signaling and metabolic stress. Increased ORP5 expression has been associated with pathological cardiac hypertrophy, suggesting that aberrant ORP5-mTORC1 interactions may drive maladaptive metabolic responses in HF [119]. These findings indicate that targeting ORP5-mediated mTORC1 activation may represent a novel therapeutic approach for mitigating myocardial stress and preserving cardiac function. Translating lysosomal-targeted strategies into clinical practice requires further investigation to determine whether pharmacological interventions can restore lysosomal integrity and improve cardiac outcomes in patients with HF. TFEB activators, TRPML1 agonists, and NLRP3 inflammasome inhibitors are among the promising candidates that may help stabilize lysosomal homeostasis and alleviate metabolic and inflammatory stress in patients with HF [72,77]. Moreover, integrating lysosomal biomarkers into HF risk stratification models could provide valuable clinical insights, enabling early intervention and personalized treatment.

Further research should aim to clarify how lysosomal dysfunction contributes to distinct HF phenotypes, particularly HF with reduced ejection fraction (HFrEF) and HF with preserved ejection fraction (HFpEF). While lysosomal impairment in HFpEF may promote metabolic cardiomyopathy by disrupting autophagic clearance and increasing myocardial stiffness, excessive lysosomal membrane permeabilization in HFrEF may exacerbate inflammatory cascades and induce cardiomyocyte apoptosis. Understanding these differential mechanisms is essential for designing targeted interventions. Incorporating lysosomal-targeted therapies into standard HF treatment regimens requires a comprehensive evaluation of their safety, efficacy, and long-term impact on cardiac function. The interplay between lysosomal dysfunction, metabolic regulation, and myocardial remodeling underscores the need for translational studies to bridge preclinical discoveries with clinical applications. Future investigations should focus on determining optimal therapeutic strategies for modulating lysosomal function in HF, ultimately paving the way for novel treatment paradigms that address the underlying pathophysiology of HF at the lysosomal level.

### 10.5. Integrating Lysosomal Dysfunction into the Broader Landscape of CVD Pathophysiology

Lysosomal stress is not an isolated phenomenon but is closely linked to other pathophysiological processes, including ferroptosis, cellular senescence, and metabolic dysregulation [91]. Future studies should explore the interplay between lysosomal function and lipid metabolism, mitochondrial homeostasis, and immune activation to identify novel therapeutic targets [92]. Additionally, emerging evidence suggests that lysosomal dysfunction may play a role in the progression of age-related CVDs, such as atherosclerosis and diabetic cardiomyopathy [93]. Understanding how lysosomal integrity declines with age and how it contributes to CVD risk could provide new avenues for therapeutic intervention [94].

Targeting lysosomal dysfunction represents a promising frontier in cardiovascular medicine. Future research should prioritize the advancement of pharmacological interventions, the development of reliable biomarkers, the translation of lysosomal therapies into clinical practice, and the integration of lysosomal dysfunction into the broader landscape of CVD pathophysiology. By addressing these challenges, lysosome-targeted strategies have the potential to redefine cardiovascular treatment paradigms and improve outcomes for patients with CVD.

**Table 2 biomedicines-13-01053-t002:** Future research avenues in lysosome-targeted cardiovascular therapies.

Category	Key Focus Areas
Advancing Lysosomal-Targeted Pharmacological Therapies	Clinical validation of TFEB activators, TRPML1 agonists, and NLRP3 inhibitors. Development of combination therapies with existing cardiovascular drugs. Optimization of dosing strategies to prevent maladaptive autophagy.
Exploring Novel Biomarkers and Diagnostic Tools	Identification of circulating lysosomal biomarkers (cathepsins, LAMP2, TFEB activity). Development of advanced imaging techniques (PET tracers, MRI-based lysosomal assessment).
Translating Strategies into Clinical Practice	Investigation of lysosomal dysfunction in heart failure subtypes (HFpEF vs. HFrEF). Evaluation of lysosomal-targeted therapies in randomized clinical trials. Integration of lysosomal biomarkers into cardiovascular risk stratification.
Expanding the Role of Lysosomal Dysfunction in CVD Pathophysiology	Examination of the interplay between lysosomal stress, ferroptosis, and cellular senescence. Investigation of lysosomal impairment in age-related cardiovascular diseases (e.g., atherosclerosis, diabetic cardiomyopathy).

## Figures and Tables

**Figure 2 biomedicines-13-01053-f002:**
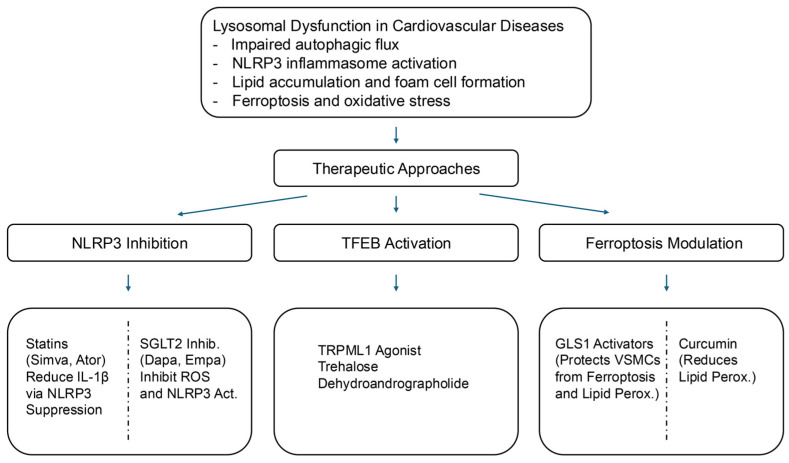
Therapeutic strategies targeting lysosomal dysfunction in cardiovascular diseases. This schematic illustrates lysosomal dysfunction as a central contributor to CVDs and highlights emerging therapeutic strategies targeting these pathways. Lysosomal stress disrupts autophagic flux, promotes lipid accumulation, activates the NLRP3 inflammasome, and triggers ferroptosis, collectively exacerbating inflammation and oxidative stress.

**Figure 3 biomedicines-13-01053-f003:**
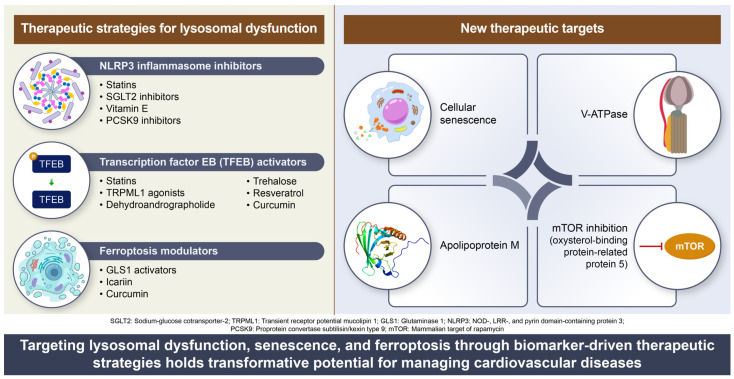
Therapeutic strategies targeting lysosomal dysfunction in CVDs. This schematic illustrates the key therapeutic strategies designed to mitigate lysosomal dysfunction in CVDs by targeting various pathological mechanisms.

**Table 1 biomedicines-13-01053-t001:** Mechanism of action and clinical status of lysosome-targeted therapies.

Drug Class	Mechanism of Action	Clinical Status	Key Clinical Findings
NLRP3 Inflammasome Inhibitors			
Statins	Inhibits NLRP3 inflammasome via AMPK activation, reducing inflammation	Clinical use	Lowers inflammatory burden in CVD
SGLT2 Inhibitors	Suppresses NLRP3 inflammasome activation enhances autophagy	Clinical use	Reduces myocardial infarction size, cardiac damage markers
Vitamin E	Inhibits Lamtor1-HDAC6 interaction, reducing NLRP3 activation	Observational and preclinical studies	Associated with reduced CVD risk
Cardiovascular Drugs Enhancing TFEB Activity			
Statins	Enhances TFEB activity via mTORC1 inhibition, AMPK activation	Widely used in clinical practice	Improves lysosomal function, reduces lipid accumulation
TRPML1 Agonists	Activates TFEB via lysosomal calcium signaling	Preclinical	Protects against oxidative stress and autophagic defects
Trehalose	Promotes TFEB activation, enhances lysosomal biogenesis	Preclinical, under investigation	Reduces plaque burden, enhances autophagy
Resveratrol (RSV)	Stimulates ER-Ca^2^⁺ signaling and activates TFEB	Preclinical evidence	Improves lipid metabolism and autophagy regulation
Curcumin (Cur)	Promotes TFEB activation, enhances lipid catabolism	Preclinical studies	Reduces foam cell formation and inflammation
Dehydroandrographolide (DA)	Activates TFEB enhances the lysosomal function	Experimental models	Improves autophagic flux, reduces CVD progression
Ferroptosis Modulators			
GLS1 Activators	Enhances glutaminolysis, reducing ferroptosis	Preclinical studies	Prevents oxidative stress-induced VSMC death
Icariin	Reduces ROS, promotes TFEB nuclear translocation, prevents ferroptosis	Preclinical models	Reduces atherosclerotic lesions

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
