# Peer review of "Lysosomal Stress in Cardiovascular Diseases: Therapeutic Potential of Cardiovascular Drugs and Future Directions"

_biomedicines, 2025, doi:10.3390/biomedicines13051053_

Round 1
Reviewer 1 Report
Comments and Suggestions for Authors
Major Comments:
1. The abstract is difficult to follow and appears disjointed. The abstract also does not appear to connect with the topics of the article. Please consider revising both so they better coordinate.
2. There appears to be areas of the article that are lacking in relevant citations to the previous work particularly with regard to the end of the article.
3. It is confusing that section 8 appears to be formatted in bullet formats rather than discussing the topic as a research article similar to the majority of the review.
4. There are section of the article ( example 5.4) that appear to be lacking portions of discussion or appear like they belong in other sections. It is unclear why the article will discuss something in sections below when the review then moves to another section of discussion.
5. The article title and discussions within section are disjointed and confusing to follow. Please consider revising so titles and the discussion below the title coordinate.
Comments on the Quality of English LanguageThe article is difficult to follow from sentence to sentence because there is no logical flow of the article. The article needs to be fully re-written before it would be acceptable for publication. The best example of the disjointed nature of the article is the abstract.
Reviewer 2 Report
Comments and Suggestions for Authors
This paper reviews the role of lysosomal dysfunction in cardiovascular disease (CVD), highlighting its relationship with ferroptosis, cellular senescence, and metabolic stress. It also discusses emerging therapeutic strategies. Key pathways such as mTORC1, TFEB, and GLS1 are analyzed, along with potential drugs such as statins, SGLT2 inhibitors, and TRPML1 activators.
1. If possible, consolidate similar ideas into a single section, e.g., the summary and future directions of sections 2, 3, and 4. Include current challenges and limitations in the development of lysosome-targeted therapies.
2. Include information on the current status of clinical trials of therapies targeting lysosomal dysfunction in CVD.
3. To compare therapeutic strategies in terms of efficacy and safety based on clinical data.
4. Many of the therapies discussed are based on preclinical studies, but human trials are rarely mentioned in the text. It is recommended that a comparative section with evidence from clinical trials be included, e.g., mentioning investigational drugs in the trial phase and their safety and efficacy in patients.
5. A table of the drugs analyzed (statins, SGLT2 inhibitors, mTOR inhibitors, TRPML1 activators, GLS1 modulators), highlighting their mechanism of action and clinical status would help to clarify the document.
6. References 6, 7, 15, 16, 17, and 50 are not formatted like the others.
Reviewer 3 Report
Comments and Suggestions for Authors
- The authors (Otoda T. et al (Lysosomal Stress in Cardiovascular Diseases: Therapeutic Potential of Cardiovascular Drugs and Future Directions) provided an interesting and original review on this stress during cardiovascular diseases.
- The review is organized in different chapters. After a short introduction on CVD, the cellular and cardiovascular lysosomal stress was introduced. Emerging therapeutic strategies was also included.
- The mechanism and therapeutic insights were presented in a second chapter. A nice and clear figure illustrated the mechanism (a similar scheme with the therapeutic ways should be helpful).
A third part focus on TFEB as a key mechanism of interaction in lysosomal stress during CVD, why this mode avec key ? The part 4 discussed the therapeutic potential of targeting lysosome stress during CVD progression. Senescence associated to cellular lysosomal dysfunction was introduced in a chapter 5.
A part 6 discussed the role of the intracellular acidic H+-ATPase with fatty acid and the balance for saturated and monounsaturated FA. Is there a role for PUFA (such as omega-3 FA of interest within CVD).
A conclusion was present with a nice figure for the therapeutic effects of some agents on lysosomal stress. (Why all the substances in the review are not presented, is there a reason, the notion of evidence ?). A conclusive figure with all the facts discussed before should be of interest for this review.
Finally future direction for lysosome stress and CVD is interesting. This is an original fact of this paper. A figure would be nice here.
- The CVD are mainly resulting from obesity, diabetes and atherosclerosis, chronic cardiomyopathy, pressure overload, heart failure show an impact of high mortality. These facts are not well explained here. A therapeutic PCSK9 for lysosomal stress, high uric acid (uremic toxins), resveratrol and curcumin (Zhao ST et al 2025 IJMS) with different signaling pathways acting as natural antioxydants (only the vitE is discussed) are linked with autophagy ferroptosis, substance like Apo lipoproteins (ApoM) ? Are they useful for the review ? A rapid Pubmed analysis on camellia sinensis (as beverage) showed no bibliographic data to the contrario.
- Many facts are preclinical data or cellular mechanism of action, the future should include a clinical approach with lysosome stress with severe CVD (heart failure).
Round 2
Reviewer 2 Report
Comments and Suggestions for Authors
Thank you for your replies.
Reviewer 3 Report
Comments and Suggestions for Authors
The present review has been revised according to reviewer's suggestions. Tables are now present and summarize the key targets of action of lysosomal dysfunction.